# Pointwise Wavelet Estimations for a Regression Model in Local Hölder Space

**Junke Kou, Qinmei Huang and Huijun Guo ***

School of Mathematics and Computational Science, Guilin University of Electronic Technology, Guilin 541004, China; kjkou@guet.edu.cn (J.K.); hqm353474937@163.com (Q.H.)
*   Correspondence: gkmath17@163.com

**Abstract:** This paper considers an unknown functional estimation problem in a regression model with multiplicative and additive noise. A linear wavelet estimator is first constructed by a wavelet projection operator. The convergence rate under the pointwise error of linear wavelet estimators is studied in local Hölder space. A nonlinear wavelet estimator is provided by the hard thresholding method in order to obtain an adaptive estimator. The convergence rate of the nonlinear estimator is the same as the linear estimator up to a logarithmic term. Finally, it should be pointed out that the convergence rates of two wavelet estimators are consistent with the optimal convergence rate on pointwise nonparametric estimation.

**Keywords:** nonparametric estimation; pointwise error; local Hölder space; wavelet

## 1. Introduction

The classical regression model plays an important role in many practical applications. The definition of this model is shown by $Y_i = f(X_i) + \varepsilon_i, i \in \{1, \ldots, n\}$. The aim of this conventional regression model is to estimate the unknown regression function $f(x)$ by observed data $(X_1, Y_1), \ldots, (X_n, Y_n)$. For this classical regression model, many important and interesting results have been obtained by Hart [1], Kerkyacharian and Picard [2], Chesneau [3], Reiß [4], Yuan and Zhou [5], and Wang and Politis [6].

Recently, Chesneau et al. [7] studied the following regression model

$$Y_i = f(X_i)U_i + V_i, i \in \{1, \ldots, n\}, \tag{1}$$

where $(X_1, Y_1), \ldots, (X_n, Y_n)$ are independent and identically distributed random variables, $f$ is an unknown function defined on $\Delta \subseteq \mathbb{R}$, $U_1, \ldots, U_n$ are $n$ identically distributed random vectors, $X_1, \ldots, X_n$ and $V_1, \ldots, V_n$ are identically distributed random variables. Moreover, $X_i$ and $U_i$ are independent, $U_i$ and $V_i$ are independent for any $i \in \{1, \ldots, n\}$. The aim of this model is to estimate the unknown function $r(x)(r := f^2)$ by the observed data $(X_1, Y_1), \ldots, (X_n, Y_n)$.

For the above model (1), it reduces to the classical regression model when $U_i \equiv 1$. In other words, (1) can be viewed as an extension of the classical regression problem. In addition, model (1) becomes the classical heteroscedastic regression model when $V_i$ is a function of $X_i$ ($V_i = g(X_i)$). Then, the function $r(x)(r := f^2)$ is called a variance function in a heteroscedastic regression model, which plays a crucial role in financial and economic fields (Cai and Wang [8], Alharbi and Patili [9]). Furthermore, the regression model (1) is also widely used in Global Positioning Systems (Huang et al. [10]), Image processing (Kravchenko et al. [11], Cui [12]), and so on.

For this regression model, Chesneau et al. [7] propose two wavelet estimators and discuss convergence rates under the mean integrated square error over Besov space. However, this study only focuses on the global error of wavelet estimators. There is a lack of pointwise risk estimation for this model. In this paper, two new wavelet estimators are constructed, and the convergence rates over the pointwise error of wavelet estimators in local Hölder space are considered. More importantly, those wavelet estimators can all obtain the optimal convergence rate under pointwise error.

## 2. Assumptions, Local Hölder Space and Wavelet

In this paper, we will consider model (1) with $\Delta = [0, 1]$. Additional technical assumptions are formulated below.

- A1:$Y_i$ is bounded for any $i \in \{1, \ldots, n\}$.
- A2:$X_1 \sim U(0, 1)$.
- A3:$U_1 \sim N(0, 1)$.
- A4:$V_1$ has a moment of order 2.
- A5:$X_i$ and $V_i$ are independent for any $i \in \{1, \ldots, n\}$.
- A6:$V_i = g(X_i)$, where $g \colon [0, 1] \to \mathbb{R}$ is known and bounded.

For the above assumptions, it is easy to see that A5 and A6 are reversed. Hence, we will define the following two sets, H1 and H2, of the above assumptions

$$H1:=\{A1,A2,A3,A4,A5\},$$
$$H2:=\{A1,A2,A3,A4,A6\}.$$

Note that the difference between H1 and H2 is the relationship between $V_i$ and $X_i$. Since the above assumptions are separated into two sets, H1 and H2; the estimators of the function $r(x)$ should be constructed under different condition sets, respectively.

This paper will consider nonparametric pointwise estimation in local Hölder space. Now, we introduce the concept of local Hölder space. Recall the classic Hölder condition $H^\delta(\mathbb{R})(0 < \delta < 1)$,

$$|f(y) - f(x)| \leq C|y - x|^\delta, x, y \in \mathbb{R}.$$

Let $\Omega_{x_0}$ be a neighborhood of $x_0 \in \mathbb{R}$ and a function space $H^\delta(\Omega_{x_0})(0 < \delta \leq 1)$ be defined as

$$H^\delta(\Omega_{x_0}) = \left\{ f \colon |f(y) - f(x)| \leq C|y - x|^\delta, x, y \in \Omega_{x_0} \right\},$$

where $C > 0$ is a fixed constant. Clearly, $f \in H^\delta(\mathbb{R})$ must be contained in $H^\delta(\Omega_{x_0})$. However, the converse does not hold.

For $s = N + \delta > 0$ with $\delta \in (0, 1]$ and $N \in \mathbb{N}$ (the nonnegative integer set), we define the local Hölder space as

$$H^s(\Omega_{x_0}) = \left\{ f \colon f^{(N)} \in H^\delta(\Omega_{x_0}) \right\}.$$

Furthermore, it follows from the definition of $H^s(\Omega_{x_0})$ that $H^s(\Omega_{x_0}) \subseteq L^2(\mathbb{R})$.

In order to construct wavelet estimators in later sections, we introduce some basic theories of wavelets.

**Definition 1.** *A multiresolution analysis (MRA) is a sequence of closed subspaces $\{V_j\}_{j \in \mathbb{Z}}$ of the square-integrable function space $L^2(\mathbb{R})$ satisfying the following properties:*
*(i) $V_j \subseteq V_{j+1}$;*
*(ii) $\overline{\bigcup_{j \in \mathbb{Z}} V_j} = L^2(\mathbb{R})$(the space $\bigcup_{j \in \mathbb{Z}} V_j$ is dense in $L^2(\mathbb{R})$);*
*(iii) $f(2\cdot) \in V_{j+1}$ if and only if $f(\cdot) \in V_j$ for each $j \in \mathbb{Z}$;*
*(iv) There exists $\phi \in L^2(\mathbb{R})$ (scaling function) such that $\{\phi(\cdot - k), k \in \mathbb{Z}\}$ forms an orthonormal basis of $V_0 = \overline{span}\{\phi(\cdot - k)\}$.*
*Let $\phi$ be a scaling function, and $\psi$ be a wavelet function such that*

$$\{\phi_{j_*,k}, \psi_{j,k}, j \geq j_*, k \in \mathbb{Z}\}$$

constitutes an orthonormal basis of $L^2(\mathbb{R})$, where $j_*$ is a positive integer, $\phi_{j_*,k} = 2^{\frac{j_*}{2}}\phi(2^{j_*}x - k)$ and $\psi_{j,k} = 2^{\frac{j}{2}}\psi(2^j x - k)$. In this paper, we choose the Daubechies wavelets. Then for any $h(x) \in H^s(\Omega_{x_0})$, it has the following expansion

$$h(x) = \sum_{k \in \mathbb{Z}} \alpha_{j_*,k} \phi_{j_*,k}(x) + \sum_{j \geq j_*} \sum_{k \in \mathbb{Z}} \beta_{j,k} \psi_{j,k}(x),$$

where $\alpha_{j,k} = \langle h, \phi_{j,k} \rangle$, $\beta_{j,k} = \langle h, \psi_{j,k} \rangle$. Further details can be found in Meyer [13] and Daubechies [14].

Let $P_j$ be the orthogonal projection operator from $L^2(\mathbb{R})$ onto the space $V_j$ with the orthonormal basis $\left\{ \phi_{j,k}(\cdot) = 2^{\frac{j}{2}}\phi(2^j \cdot -k), k \in \mathbb{Z} \right\}$. Then for $h(x) \in H^s(\Omega_{x_0})$ and $\alpha_{j,k} = \langle h, \phi_{j,k} \rangle$,

$$P_j h(x) = \sum_{k \in \mathbb{Z}} \alpha_{j,k} \phi_{j,k}(x).$$

In this position, we give an important lemma, which will be used in later discussions. Here and after, we adopt the following symbol: $A \lesssim B$ denotes $A \leq cB$ for some constant $c > 0$; $A \gtrsim B$ means $B \lesssim A$; $A \sim B$ stand for both $A \lesssim B$ and $B \lesssim A$.

**Lemma 1** (Liu and Wu [15]). *If $f \in H^s(\Omega_{x_0})$, $s > 0$ with $s = N + \delta (0 < \delta \leq 1)$, then for $x \in \Omega_{x_0}$ and $j_* \in \mathbb{N}$,*

*(i)* $\displaystyle\sup_{f \in H^s(\Omega_{x_0})} \sum_{k \in \mathbb{Z}} \left| \beta_{j,k} \psi_{j,k}(x) \right| \lesssim 2^{-js};$

*(ii)* $f(x) = \displaystyle\sum_{k \in \mathbb{Z}} \alpha_{j_*,k} \phi_{j_*,k}(x) + \sum_{j \geq j_*} \sum_{k \in \mathbb{Z}} \beta_{j,k} \psi_{j,k};$

*(iii)* $\displaystyle\sup_{f \in H^s(\Omega_{x_0})} \left| f(x) - P_{j_*} f(x) \right| \lesssim 2^{-j_* s}.$

## 3. Linear Wavelet Estimator

In this section, a linear wavelet estimator is given by using the wavelet method, and the order of pointwise convergence of this estimator is studied in local Hölder space. Now we define our linear wavelet estimator

$$\hat{r}_n^{lin}(x) = \sum_k \hat{\alpha}_{j_*,k} \phi_{j_*,k}(x), \tag{2}$$

where

$$\hat{\alpha}_{j_*,k} = \frac{1}{n} \sum_{i=1}^n Y_i^2 \phi_{j_*,k}(X_i) - v_{j_*,k}, \tag{3}$$

$$v_{j_*,k} = \begin{cases} \mathbb{E}[V_1^2]2^{-j_*/2}, & \text{A5,} \\ \int_0^1 g^2(x)\phi_{j_*,k}(x)dx, & \text{A6.} \end{cases} \tag{4}$$

According to the definition of $v_{j_*,k}$, it is clear that the structure of this linear wavelet estimator depends on the reverse conditions of A5 and A6. Some of the lemmas needed in this section and their proofs are given below.

**Lemma 2.** *For model (1), if H1 or H2 hold,*

$$\mathbb{E}[\hat{\alpha}_{j_*,k}] = \alpha_{j_*,k}. \tag{5}$$

**Proof.** According to the definition of $\hat{\alpha}_{j_*,k}$,

$$\mathbb{E}[\hat{\alpha}_{j_*,k}] = \mathbb{E}\left[\frac{1}{n}\sum_{i=1}^{n} Y_i^2 \phi_{j_*,k}(X_i) - v_{j_*,k}\right]$$

$$= \mathbb{E}\left[Y_1^2 \phi_{j_*,k}(X_1)\right] - v_{j_*,k}$$

$$= \mathbb{E}\left[r(X_1)U_1^2 \phi_{j_*,k}(X_1)\right] + 2\mathbb{E}[f(X_1)U_1 V_1 \phi_{j_*,k}(X_1)] + \mathbb{E}\left[V_1^2 \phi_{j_*,k}(X_1)\right] - v_{j_*,k}.$$

Since $U_i$ is independent from $X_i$ and $V_i$, respectively,

$$\mathbb{E}[f(X_1)U_1 V_1 \phi_{j_*,k}(X_1)] = \mathbb{E}[U_1]\mathbb{E}[f(X_1)V_1 \phi_{j_*,k}(X_1)].$$

In addition, condition A3 implies that $\mathbb{E}[U_1] = 0$. Then one gets

$$\mathbb{E}[f(X_1)U_1 V_1 \phi_{j_*,k}(X_1)] = 0.$$

It follows from A5, A2 and A4 that

$$\mathbb{E}[V_1^2]\mathbb{E}[\phi_{j_*,k}(X_1)] = \mathbb{E}\left[V_1^2\right]\int_0^1 \phi_{j_*,k}(x)dx = \mathbb{E}[V_1^2]2^{-\frac{j_*}{2}} = v_{j_*,k}.$$

On the other hand, we obtain

$$\mathbb{E}\left[V_1^2 \phi_{j_*,k}(X_1)\right] = \int_0^1 g^2(x)\phi_{j_*,k}(x)dx = v_{j_*,k}$$

with condition A6.

Finally, according to the assumption of A3 and A2,

$$\mathbb{E}[\hat{\alpha}_{j_*,k}] = \mathbb{E}[U_1^2]\mathbb{E}[r(X_1)\phi_{j_*,k}(X_1)] = \int_0^1 r(x)\phi_{j_*,k}(x)dx = \alpha_{j_*,k}.$$

$\square$

In order to estimate $\mathbb{E}\left[\left|\hat{\alpha}_{j_*,k} - \alpha_{j_*,k}\right|^p\right]$, we need the following Rosenthal's inequality.

**Rosenthal's inequality** Let $X_1, \ldots, X_n$ be independent random variables such that $\mathbb{E}[X_i] = 0$ and $|X_i| \leq M (i = 1, 2, \ldots, n)$,

(i) $\mathbb{E}\left[\left|\sum_{i=1}^{n} X_i\right|^p\right] \lesssim \left(M^{p-2}\sum_{i=1}^{n}\mathbb{E}[X_i^2] + \left(\sum_{i=1}^{n}\mathbb{E}[X_i^2]\right)^{p/2}\right), p > 2;$

(ii) $\mathbb{E}\left[\left|\sum_{i=1}^{n} X_i\right|^p\right] \lesssim \left(\sum_{i=1}^{n}\mathbb{E}[X_i^2]\right)^{p/2}, 0 < p \leq 2.$

**Lemma 3.** *Let $\hat{\alpha}_{j_*,k}$ be defined by* (3). *If H1 or H2 hold and $2^{j_*} \leq n$, then for $1 \leq p < \infty$,*

$$\mathbb{E}\left[\left|\hat{\alpha}_{j_*,k} - \alpha_{j_*,k}\right|^p\right] \lesssim n^{-p/2}. \tag{6}$$

**Proof.** By (5) and the definition of $\hat{\alpha}_{j_*,k}$,

$$|\hat{\alpha}_{j_*,k} - \alpha_{j_*,k}| = \left|\frac{1}{n}\sum_{i=1}^{n} Y_i^2 \phi_{j_*,k}(X_i) - v_{j_*,k} - \mathbb{E}\left[\frac{1}{n}\sum_{i=1}^{n} Y_i^2 \phi_{j_*,k}(X_i) - v_{j_*,k}\right]\right|$$

$$= \frac{1}{n}\left|\sum_{i=1}^{n}(Y_i^2 \phi_{j_*,k}(X_i) - \mathbb{E}\left[Y_i^2 \phi_{j_*,k}(X_i)\right])\right| = \frac{1}{n}\left|\sum_{i=1}^{n} Z_i\right| \tag{7}$$

with $Z_i := Y_i^2 \phi_{j_*,k}(X_i) - \mathbb{E}[Y_i^2 \phi_{j_*,k}(X_i)]$. It is clear that $\mathbb{E}[Z_i] = 0$. Using the definition of $Z_i$ and A1, there exists a constant $c > 0$ such that

$$|Z_i| = \left| Y_i^2 \phi_{j_*,k}(X_i) - \mathbb{E}\left[ Y_i^2 \phi_{j_*,k}(X_i) \right] \right| \le \left| Y_i^2 \phi_{j_*,k}(X_i) \right| + \left| \mathbb{E}\left[ Y_i^2 \phi_{j_*,k}(X_i) \right] \right| \le c 2^{\frac{j_*}{2}} \lesssim 2^{\frac{j_*}{2}}.$$

When $p > 2$, according to Rosenthal's inequality,

$$\mathbb{E}\left[ \left| \sum_{i=1}^n Z_i \right|^p \right] \lesssim \left( M^{p-2} \sum_{i=1}^n \mathbb{E}\left[ Z_i^2 \right] + \left( \sum_{i=1}^n \mathbb{E}\left[ Z_i^2 \right] \right)^{\frac{p}{2}} \right)$$

$$\lesssim (2^{\frac{j_*}{2}})^{p-2} \sum_{i=1}^n \mathbb{E}\left[ Z_i^2 \right] + \left( \sum_{i=1}^n \mathbb{E}\left[ Z_i^2 \right] \right)^{\frac{p}{2}}. \tag{8}$$

Note that $\mathbb{E}[Z_i^2] = Var[Z_i] = Var\left[ Y_i^2 \phi_{j_*,k}(X_i) - \mathbb{E}[Y_i^2 \phi_{j_*,k}(X_i)] \right] = Var\left[ Y_i^2 \phi_{j_*,k}(X_i) \right] \le \mathbb{E}\left[ Y_i^4 \phi_{j_*,k}^2(X_i) \right]$. Furthermore, it follows from A1 and the property of $\phi_{j_*,k}$ that

$$\mathbb{E}\left[ Z_i^2 \right] \lesssim \mathbb{E}\left[ Y_i^4 \phi_{j_*,k}^2(X_i) \right] \lesssim 1.$$

Then it can be easily seen that

$$\left( \sum_{i=1}^n \mathbb{E}\left[ Z_i^2 \right] \right)^{p/2} \lesssim n^{\frac{p}{2}}. \tag{9}$$

By (8) and (9), we obtain

$$\mathbb{E}\left[ \left| \sum_{i=1}^n Z_i \right|^p \right] \lesssim (2^{\frac{j_*}{2}})^{p-2} n + n^{\frac{p}{2}}. \tag{10}$$

When $1 \le p < 2$,

$$\mathbb{E}\left[ \left| \sum_{i=1}^n Z_i \right|^p \right] \lesssim \left( \sum_{i=1}^n \mathbb{E}\left[ Z_i^2 \right] \right)^{p/2}.$$

Hence,

$$\mathbb{E}\left[ \left| \sum_{i=1}^n Z_i \right|^p \right] \lesssim n^{\frac{p}{2}}. \tag{11}$$

It follows from (7), (10) and (11) that

$$\mathbb{E}\left[ |\hat{\alpha}_{j_*,k} - \alpha_{j_*,k}|^p \right] \lesssim \mathbb{E}\left[ \left( \frac{1}{n} \left| \sum_{i=1}^n Z_i \right| \right)^p \right] = \frac{1}{n^p} \mathbb{E}\left[ \left| \sum_{i=1}^n Z_i \right|^p \right].$$

Hence,

$$\mathbb{E}\left[ |\hat{\alpha}_{j_*,k} - \alpha_{j_*,k}|^p \right] \lesssim \begin{cases} \frac{1}{n^p}\left[ (2^{\frac{j_*}{2}})^{p-2} \cdot n + n^{\frac{p}{2}} \right], & p \ge 2, \\ n^{-\frac{p}{2}}, & 1 \le p < 2. \end{cases} \tag{12}$$

This with $2^{j_*} \le n$ implies that

$$\mathbb{E}[|\hat{\alpha}_{j_*,k} - \alpha_{j_*,k}|^p] \lesssim n^{-\frac{p}{2}}.$$

□

Now the convergence rate of the linear wavelet estimator is proved in the following.

**Theorem 1.** *Let* $r \in H^s(\Omega_{x_0})$ *with* $s > 0$. *Then for each* $1 \leq p < \infty$, *the linear wavelet estimator* $\hat{r}_n^{lin}(x)$ *defined in* (2) *with* $2^{j_*} \sim n^{\frac{1}{2s+1}}$ *satisfies*

$$\sup_{r \in H^s(\Omega_{x_0})} \left\{ \mathbb{E}\left[ |\hat{r}_n^{lin}(x_0) - r(x_0)|^p \right] \right\}^{\frac{1}{p}} \lesssim n^{-\frac{s}{2s+1}}.$$

**Remark 1.** *Note that* $n^{-\frac{s}{2s+1}}$ *is the optimal convergence rate over pointwise error for nonparametric functional estimation (Brown and Low [16]). The above result yields that the linear wavelet estimator can obtain the optimal convergence rate.*

**Proof.** The triangular inequality gives

$$\left\{ \mathbb{E}\left[ \left| \hat{r}_n^{lin}(x_0) - r(x_0) \right|^p \right] \right\}^{\frac{1}{p}} \lesssim \left\{ \mathbb{E}\left[ \left| \hat{r}_n^{lin}(x_0) - P_{j_*}r(x_0) \right|^p + \left| P_{j_*}r(x_0) - r(x_0) \right|^p \right] \right\}^{\frac{1}{p}}$$

$$\lesssim \left\{ \mathbb{E}\left[ \left| \hat{r}_n^{lin}(x_0) - P_{j_*}r(x_0) \right|^p \right] \right\}^{\frac{1}{p}}$$
$$+ \left| P_{j_*}r(x_0) - r(x_0) \right|. \tag{13}$$

- The bias term $\left| P_{j_*}r(x_0) - r(x_0) \right|$. According to Lemma 1,

$$\left| P_{j_*}r(x_0) - r(x_0) \right| \lesssim 2^{-j_*s}. \tag{14}$$

- The stochastic term $\left\{ \mathbb{E}\left[ \left| \hat{r}_n^{lin}(x_0) - P_{j_*}r(x_0) \right|^p \right] \right\}^{\frac{1}{p}}$. Note that

$$\mathbb{E}\left[ \left| \hat{r}_n^{lin}(x_0) - P_{j_*}r(x_0) \right|^p \right] = \mathbb{E}\left[ \left| \sum_{k \in \Lambda_{j_*}} \left( \hat{\alpha}_{j_*,k} - \alpha_{j_*,k} \right) \phi_{j_*,k}(x_0) \right|^p \right]$$

$$\leq \mathbb{E}\left[ \left\{ \sum_{k \in \Lambda_{j_*}} \left| \hat{\alpha}_{j_*,k} - \alpha_{j_*,k} \right| \left| \phi_{j_*,k}(x_0) \right|^{\frac{1}{p}} \left| \phi_{j_*,k}(x_0) \right|^{\frac{1}{p'}} \right\}^p \right]$$

with $\frac{1}{p} + \frac{1}{p'} = 1$. According to the Hölder inequality, Lemma 3 and $\sum_{k \in \Lambda_{j_*}} \left| \phi_{j_*,k} \right| \lesssim 2^{j_*/2}$, the above inequality reduces to

$$\mathbb{E}\left[ \left| \hat{r}_n^{lin}(x_0) - P_{j_*}r(x_0) \right|^p \right]$$

$$\leq \mathbb{E}\left[ \left\{ \left( \sum_{k \in \Lambda_{j_*}} \left| \hat{\alpha}_{j_*,k} - \alpha_{j_*,k} \right|^p \left| \phi_{j_*,k}(x_0) \right| \right)^{\frac{1}{p}} \left( \sum_{k \in \Lambda_{j_*}} \left| \phi_{j_*,k}(x_0) \right| \right)^{\frac{1}{p'}} \right\}^p \right]$$

$$\lesssim \sum_{k \in \Lambda_{j_*}} \mathbb{E}\left[ \left| \hat{\alpha}_{j_*,k} - \alpha_{j_*,k} \right|^p \left| \phi_{j_*,k}(x_0) \right| \right] 2^{\frac{j_* p}{2p'}}$$

$$\lesssim \left( \frac{1}{n} \right)^{\frac{p}{2}} 2^{\frac{j_*}{2}\left(1+\frac{p}{p'}\right)} = \left( \frac{2^{j_*}}{n} \right)^{\frac{p}{2}} \tag{15}$$

Combining (13), (14) and (15), one has

$$\left\{ \mathbb{E}\left[ \left| \hat{r}_n^{lin}(x_0) - r(x_0) \right|^p \right] \right\}^{1/p} \leq 2^{-j_*s} + \left( \frac{2^{j_*}}{n} \right)^{\frac{1}{2}}.$$

Furthermore, by the given choice $2^{j_*} \sim n^{\frac{1}{2s+1}}$,

$$\sup_{r \in H^s(\Omega_{x_0})} \left\{ \mathbb{E}\left[ |\hat{r}_n^{lin}(x_0) - r(x_0)|^p \right] \right\}^{\frac{1}{p}} \lesssim n^{-\frac{s}{2s+1}}.$$

□

## 4. Nonlinear Wavelet Estimator

According to the definition of the linear wavelet estimator, we can easily find that the scale parameter $j_*$ of the linear wavelet estimator depends on the smooth parameter $s$ of the function $r(x)$ to be estimated, so the linear estimator is not adaptive. In this section, we will solve this problem by constructing a nonlinear wavelet estimator with the hard thresholding method.

Now we define our nonlinear wavelet estimator

$$\hat{r}_n^{non}(x) = \sum_{k \in \Lambda_{j_*}} \hat{\alpha}_{j_*,k} \phi_{j_*,k}(x) + \sum_{j=j_*}^{j_1} \sum_{k \in \Lambda_j} \hat{\beta}_{j,k} I_{\{|\hat{\beta}_{j,k}| \geq \kappa t_n\}} \psi_{j,k}(x), x \in [0,1], \tag{16}$$

where $\hat{\alpha}_{j_*,k}$ is defined by (3),

$$\hat{\beta}_{j,k} := \frac{1}{n} \sum_{i=1}^{n} Y_i^2 \psi_{j,k}(X_i) - w_{j,k}, \tag{17}$$

$$w_{j,k} := \begin{cases} 0, & \text{A5,} \\ \int_0^1 g^2(x) \psi_{j,k}(x) dx, & \text{A6,} \end{cases} \tag{18}$$

and $t_n = \sqrt{\ln n / n}$, $I_G$ denotes the indicator function over an event G. The positive integer $j_*, j_1$, and $\kappa$ will be given in Theorem 2.

**Remark 2.** *Compared with the structure of $\hat{\beta}_{j,k}$ in Chesneau et al. [7], the definition of $\hat{\beta}_{j,k}$ in this paper does not need a thresholding algorithm. In other words, this paper reduces the complexity of the nonlinear wavelet estimator.*

**Lemma 4.** *For model (1), if H1 or H2 hold, then*

$$\mathbb{E}\left[ \hat{\beta}_{j,k} \right] = \beta_{j,k}.$$

**Lemma 5.** *Let $\hat{\beta}_{j,k}$ be defined by (17). If H1 or H2 hold and $2^j \leq n$, then for $1 \leq p < \infty$,*

$$\mathbb{E}\left[ |\hat{\beta}_{j,k} - \beta_{j,k}|^p \right] \lesssim n^{-p/2}.$$

The proof methods of Lemmas 4 and 5 are similar to that of Lemmas 2 and 3, so the proofs are omitted here. For nonlinear wavelet estimation, Bernstein's inequality plays a crucial role.

**Bernstein's inequality** Let $X_1, \ldots, X_n$ be independent random variables such that $\mathbb{E}[X_i] = 0$, $|X_i| \leq M$ and $\mathbb{E}[X_i^2] = \sigma^2$, then for each $v > 0$

$$\mathbb{P}\left( \frac{1}{n} \left| \sum_{i=1}^{n} X_i \right| \geq v \right) \leq 2 \exp\left\{ -\frac{nv^2}{2(\sigma^2 + \frac{vM}{3})} \right\}.$$

**Lemma 6.** *Let* $\hat{\beta}_{j,k}$ *be defined by* (17), $t_n = \sqrt{\frac{\ln n}{n}}$ *and* $2^j \leq \frac{n}{\ln n}$. *If H1 or H2 hold, then for each* $w > 0$, *there exists a constant* $\kappa > 1$ *such that*

$$\mathbb{P}(|\hat{\beta}_{j,k} - \beta_{j,k}| \geq \kappa t_n) \lesssim 2^{-wj}.$$

**Proof.** According to the definition of $\hat{\beta}_{j,k}$,

$$
\begin{aligned}
|\hat{\beta}_{j,k} - \beta_{j,k}| &= \left| \frac{1}{n}\sum_{i=1}^{n} Y_i^2 \psi_{j,k}(X_i) - w_{j,k} - \mathbb{E}\left[ \frac{1}{n}\sum_{i=1}^{n} Y_i^2 \psi_{j,k}(X_i) - w_{j,k} \right] \right| \\
&= \left| \frac{1}{n}\sum_{i=1}^{n} Y_i^2 \psi_{j,k}(X_i) - \mathbb{E}\left[ \frac{1}{n}\sum_{i=1}^{n} Y_i^2 \psi_{j,k}(X_i) \right] \right| \\
&= \frac{1}{n}\left| \sum_{i=1}^{n} \left( Y_i^2 \psi_{j,k}(X_i) - \mathbb{E}\left[ Y_i^2 \psi_{j,k}(X_i) \right] \right) \right| = \frac{1}{n}\left| \sum_{i=1}^{n} D_i \right|
\end{aligned}
$$

with $D_i = Y_i^2 \psi_{j,k}(X_i) - \mathbb{E}[Y_i^2 \psi_{j,k}(X_i)]$. Clearly, $\mathbb{E}[D_i] = 0$. Furthermore, by A1 and the property of $\psi_{j,k}$, $\mathbb{E}[D_i^2] = Var[D_i] \leq \mathbb{E}[Y_i^4 \psi_{j,k}^2(X_i)] \lesssim 1$ and $|D_i| \leq 2^{\frac{j}{2}}$.

Note that

$$\left\{ |\hat{\beta}_{j,k} - \beta_{j,u}| \geq \kappa t_n \right\} \subseteq \left\{ \frac{1}{n}\left| \sum_{i=1}^{n} D_i \right| \geq \kappa t_n \right\}.$$

Hence,

$$\mathbb{P}(|\hat{\beta}_{j,k} - \beta_{j,k}| \geq \kappa t_n) \leq \mathbb{P}\left( \frac{1}{n}\left| \sum_{i=1}^{n} D_i \right| \geq \kappa t_n \right).$$

Using Bernstein's inequality, $t_n = \sqrt{\frac{\ln n}{n}}$ and $2^j \leq \frac{n}{\ln n}$,

$$\mathbb{P}\left( \frac{1}{n}\left| \sum_{i=1}^{n} D_i \right| \geq \kappa t_n \right) \lesssim \exp\left\{ -\frac{n(\kappa t_n)^2}{2(1 + \frac{\kappa t_n 2^{j/2}}{3})} \right\} \lesssim \exp\left\{ -\frac{\kappa^2 \ln n}{2(1 + \frac{\kappa}{3})} \right\}.$$

Then one chooses a large enough $\kappa > 1$ such that

$$\mathbb{P}(|\hat{\beta}_{j,k} - \beta_{j,k}| \geq \kappa t_n) \leq \mathbb{P}\left( \frac{1}{n}\left| \sum_{i=1}^{n} D_i \right| \geq \kappa t_n \right) \lesssim 2^{-wj}.$$

$\square$

**Theorem 2.** *Let* $r \in H^s(\Omega_{x_0})$ *with* $s > 0$. *Then for each* $1 \leq p < \infty$, *the nonlinear wavelet estimator* $\hat{r}_n^{non}(x)$ *defined in* (16) *with* $2^{j*} \sim n^{\frac{1}{2m+1}}$ ($s < m$) *and* $2^{j_1} \sim \frac{n}{\ln n}$ *satisfies*

$$\sup_{r \in H^s(\Omega_{x_0})} \left\{ \mathbb{E}[|\hat{r}_n^{non}(x_0) - r(x_0)|^p] \right\}^{\frac{1}{p}} \lesssim (\ln n)^{1 - \frac{1}{p}} \left( \frac{\ln n}{n} \right)^{s/(2s+1)}. \tag{19}$$

**Remark 3.** *Compared with the linear wavelet estimator, the nonlinear wavelet estimator does not depend on the smooth parameter of* $r(x)$. *Hence, the nonlinear estimator is adaptive. More importantly, the nonlinear estimator can also achieve the optimal convergence rate up to an* $\ln n$ *factor.*

**Proof.** By the definition of $\hat{r}_n^{lin}(x)$ and $\hat{r}_n^{non}(x)$, one has

$$\hat{r}_n^{non}(x_0) - r(x_0) = [\hat{r}_n^{lin}(x_0) - P_{j_*}r(x_0)] - [r(x_0) - P_{j_1+1}r(x_0)]$$
$$+ \sum_{j=j_*}^{j_1} \sum_{k\in\Lambda_j} \left(\hat{\beta}_{j,k}I_{\{|\hat{\beta}_{j,k}|\geq\kappa t_n\}} - \beta_{j,k}\right)\psi_{j,k}(x_0).$$

Hence,

$$\left\{\mathbb{E}[|\hat{r}_n^{non}(x_0) - r(x_0)|^p]\right\}^{\frac{1}{p}} \lesssim T_1 + T_2 + Q,$$

where

$$T_1 = \left\{\mathbb{E}\left[\left|\hat{r}_n^{lin}(x_0) - P_{j_*}r(x_0)\right|^p\right]\right\}^{\frac{1}{p}},$$

$$T_2 = \left|P_{j_1+1}r(x_0) - r(x_0)\right|,$$

$$Q = \left\{\mathbb{E}\left[\left(\sum_{j=j_*}^{j_1}\sum_{k\in\Lambda_j}\left|\left(\hat{\beta}_{j,k}I_{\{|\hat{\beta}_{j,k}|\geq\kappa t_n\}} - \beta_{j,k}\right)\psi_{j,k}(x_0)\right|\right)^p\right]\right\}^{\frac{1}{p}}.$$

- For $T_1$. It follows from (15) and $2^{j_*} \sim n^{\frac{1}{2m+1}}$ $(s < m)$ that

$$T_1 = \left\{\mathrm{E}\left[\left|\hat{r}_n^{lin}(x_0) - P_{j_*}r(x_0)\right|^p\right]\right\}^{\frac{1}{p}} \lesssim \left(\frac{2^{j_*}}{n}\right)^{1/2} \lesssim n^{-\frac{m}{2m+1}} < n^{-\frac{s}{2s+1}}. \qquad (20)$$

- For $T_2$. Using Lemma 1 and $2^{j_1} \sim \frac{n}{\ln n}$, one gets

$$T_2 = \left|P_{j_1+1}r(x_0) - r(x_0)\right| \lesssim 2^{-j_1 s} \lesssim \left(\frac{\ln n}{n}\right)^s < \left(\frac{\ln n}{n}\right)^{\frac{s}{2s+1}}. \qquad (21)$$

Then equality (19) will be proven if we can show

$$Q \lesssim (\ln n)^{1-\frac{1}{p}}\left(\frac{\ln n}{n}\right)^{s/(2s+1)}.$$

According to Hölder inequality,

$$Q \lesssim \left\{(j_1 - j_* + 1)^{p-1}\sum_{j=j_*}^{j_1}\mathrm{E}\left[\left(\sum_{k\in\Lambda_j}\left|\left(\hat{\beta}_{j,k}I_{\{|\hat{\beta}_{j,k}|\geq\kappa t_n\}} - \beta_{j,k}\right)\psi_{j,k}(x_0)\right|\right)^p\right]\right\}^{1/p}.$$

It is obvious that

$$|\hat{\beta}_{j,k}I_{\{|\hat{\beta}_{j,k}|\geq\kappa t_n\}} - \beta_{j,k}| = |\hat{\beta}_{j,k} - \beta_{j,k}|\left[I_{\{|\hat{\beta}_{j,k}|\geq\kappa t_n,|\beta_{j,k}|<\frac{\kappa t_n}{2}\}} + I_{\{|\hat{\beta}_{j,k}|\geq\kappa t_n,|\beta_{j,k}|\geq\frac{\kappa t_n}{2}\}}\right]$$
$$+ |\beta_{j,k}|\left[I_{\{|\hat{\beta}_{j,k}|<\kappa t_n,|\beta_{j,k}|>2\kappa t_n\}} + I_{\{|\hat{\beta}_{j,k}|<\kappa t_n,|\beta_{j,k}|\leq 2\kappa t_n\}}\right].$$

Moreover,

$$\left\{|\hat{\beta}_{j,k}|\geq\kappa t_n,|\beta_{j,k}|<\frac{\kappa t_n}{2}\right\} \subseteq \left\{|\hat{\beta}_{j,k}-\beta_{j,k}|>\frac{\kappa t_n}{2}\right\},$$

$$\left\{|\hat{\beta}_{j,k}|<\kappa t_n,|\beta_{j,k}|>2\kappa t_n\right\} \subseteq \left\{|\hat{\beta}_{j,k}-\beta_{j,k}|>\frac{\kappa t_n}{2}\right\},$$

$$|\hat{\beta}_{j,k}-\beta_{j,k}| \geq |\beta_{j,k}| - |\hat{\beta}_{j,k}| \geq \frac{\kappa t_n}{2}.$$

Hence, one can obtain that

$$Q \lesssim (j_1 - j_* + 1)^{1 - \frac{1}{p}} (Q_1 + Q_2 + Q_3),$$

where

$$Q_1 = \left\{ \sum_{j=j_*}^{j_1} \mathbb{E}\left[ \left( \sum_{k \in \Lambda_j} \left| \hat{\beta}_{j,k} - \beta_{j,k} \right| I_{\left\{ |\hat{\beta}_{j,k} - \beta_{j,k}| > \frac{\kappa t_n}{2} \right\}} \left| \psi_{j,k}(x_0) \right| \right)^p \right] \right\}^{1/p},$$

$$Q_2 = \left\{ \sum_{j=j_*}^{j_1} \mathbb{E}\left[ \left( \sum_{k \in \Lambda_j} \left| \hat{\beta}_{j,k} - \beta_{j,k} \right| I_{\left\{ |\beta_{j,k}| \geq \frac{\kappa t_n}{2} \right\}} \left| \psi_{j,k}(x_0) \right| \right)^p \right] \right\}^{1/p},$$

$$Q_3 = \sum_{j=j_*}^{j_1} \sum_{k \in \Lambda_j} \left| \beta_{j,k} \right| I_{\left\{ |\beta_{j,k}| \leq 2\kappa t_n \right\}} \left| \psi_{j,k}(x_0) \right|.$$

- For $Q_1$. By Hölder inequality ($\frac{1}{p} + \frac{1}{p'} = 1$) and $\sum_k \left| \psi_{j,k}(x_0) \right| \lesssim 2^{j/2}$

$$\mathbb{E}\left[ \left( \sum_{k \in \Lambda_j} \left| \hat{\beta}_{j,k} - \beta_{j,k} \right| I_{\left\{ |\hat{\beta}_{j,k} - \beta_{j,k}| > \frac{\kappa t_n}{2} \right\}} \left| \psi_{j,k}(x_0) \right| \right)^p \right]$$

$$= \mathbb{E}\left[ \left( \sum_{k \in \Lambda_j} \left| \hat{\beta}_{j,k} - \beta_{j,k} \right| I_{\left\{ |\hat{\beta}_{j,k} - \beta_{j,k}| > \frac{\kappa t_n}{2} \right\}} \left| \psi_{j,k}(x_0) \right|^{1/p} \left| \psi_{j,k}(x_0) \right|^{1/p'} \right)^p \right]$$

$$\leq \mathbb{E}\left[ \sum_{k \in \Lambda_j} \left| \hat{\beta}_{j,k} - \beta_{j,k} \right|^p I_{\left\{ |\hat{\beta}_{j,k} - \beta_{j,k}| > \frac{\kappa t_n}{2} \right\}} \left| \psi_{j,k}(x_0) \right| \right] \left( \sum_k \left| \psi_{j,k}(x_0) \right| \right)^{p/p'}$$

$$\leq \mathbb{E}\left[ \sum_{k \in \Lambda_j} \left| \hat{\beta}_{j,k} - \beta_{j,k} \right|^p I_{\left\{ |\hat{\beta}_{j,k} - \beta_{j,k}| > \frac{\kappa t_n}{2} \right\}} \left| \psi_{j,k}(x_0) \right| \right] 2^{\frac{jp}{2p'}}. \tag{22}$$

Furthermore, using the Cauchy–Schwarz inequality, Lemmas 5 and 6, one has

$$\mathbb{E}\left[ \left| \hat{\beta}_{j,k} - \beta_{j,k} \right|^p I_{\left\{ |\hat{\beta}_{j,k} - \beta_{j,k}| > \frac{\kappa t_n}{2} \right\}} \right]$$

$$\leq \left( \mathbb{E}\left[ \left| \hat{\beta}_{j,k} - \beta_{j,k} \right|^{2p} \right] \right)^{1/2} \left( \mathbb{E}\left[ I_{\left\{ |\hat{\beta}_{j,k} - \beta_{j,k}| > \frac{\kappa t_n}{2} \right\}} \right] \right)^{1/2} \lesssim n^{-\frac{p}{2}} 2^{-\frac{wj}{2}}. \tag{23}$$

This with (22) yields that

$$\mathbb{E}\left[ \left( \sum_{k \in \Lambda_j} \left| \hat{\beta}_{j,k} - \beta_{j,k} \right| I_{\left\{ |\hat{\beta}_{j,k} - \beta_{j,k}| > \frac{\kappa t_n}{2} \right\}} \left| \psi_{j,k}(x_0) \right| \right)^p \right]$$

$$\lesssim 2^{\frac{jp}{2}} \mathbb{E}\left[ \left| \hat{\beta}_{j,k} - \beta_{j,k} \right|^p I_{\left\{ |\hat{\beta}_{j,k} - \beta_{j,k}| > \frac{\kappa t_n}{2} \right\}} \right] \lesssim n^{-\frac{p}{2}} 2^{-\frac{wj}{2}} 2^{\frac{jp}{2}}. \tag{24}$$

Hence,

$$Q_1 \lesssim \left( \sum_{j=j_*}^{j_1} 2^{\frac{jp}{2}} n^{-\frac{p}{2}} 2^{-\frac{wj}{2}} \right)^{\frac{1}{p}} = \left( n^{-\frac{p}{2}} \sum_{j=j_*}^{j_1} 2^{j\left( \frac{p}{2} - \frac{w}{2} \right)} \right)^{\frac{1}{p}} \lesssim \left( n^{-\frac{p}{2}} 2^{j_* \frac{p}{2}} \right)^{\frac{1}{p}} = \left( \frac{2^{j_*}}{n} \right)^{\frac{1}{2}},$$

where $\kappa$ is chosen to be large enough such that $w > p$ in Lemma 6. This with the choice $2^{j_*} \sim n^{\frac{1}{2m+1}} (s < m)$ shows that

$$Q_1 \lesssim n^{-\frac{m}{2m+1}} \lesssim n^{-\frac{s}{2s+1}}. \tag{25}$$

- For $Q_2$. Let us first define

$$2^{j'} \sim \left(\frac{n}{\ln n}\right)^{1/(2s+1)}.$$

Clearly, $2^{j_*} \sim n^{\frac{1}{2m+1}} \le 2^{j'} \sim \left(\frac{n}{\ln n}\right)^{1/(2s+1)} \le 2^{j_1} \sim \frac{n}{\ln n}$. Note that

$$Q_2 = \left\{\sum_{j=j_*}^{j_1} \mathbb{E}\left[\left(\sum_{k \in \Lambda_j} \left|\hat{\beta}_{j,k} - \beta_{j,k}\right| I_{\{|\beta_{j,k}| \ge \frac{\kappa t_n}{2}\}} \left|\psi_{j,k}(x_0)\right|\right)^p\right]\right\}^{1/p}$$

$$\le \left\{\sum_{j=j_*}^{j'} \mathbb{E}\left[\left(\sum_{k \in \Lambda_j} \left|\hat{\beta}_{j,k} - \beta_{j,k}\right|\left|\psi_{j,k}(x_0)\right|\right)^p\right]\right\}^{1/p}$$

$$+ \left\{\sum_{j=j'+1}^{j_1} \mathbb{E}\left[\left(\sum_{k \in \Lambda_j} \left|\hat{\beta}_{j,k} - \beta_{j,k}\right|\frac{\left|\beta_{j,k}\right|}{t_n}\left|\psi_{j,k}(x_0)\right|\right)^p\right]\right\}^{1/p}.$$

Similar to the argument of (15), one gets

$$\left\{\sum_{j=j_*}^{j'} \mathbb{E}\left[\left(\sum_{k \in \Lambda_j} \left|\hat{\beta}_{j,k} - \beta_{j,k}\right|\left|\psi_{j,k}(x_0)\right|\right)^p\right]\right\}^{1/p} \lesssim \left(\sum_{j=j_*}^{j'} n^{-\frac{p}{2}} 2^{\frac{jp}{2}}\right)^{\frac{1}{p}} \lesssim \left(\frac{2^{j'}}{n}\right)^{1/2}. \tag{26}$$

On the other hand, by Hölder inequality ($\frac{1}{p} + \frac{1}{p'} = 1$) and Lemma 1

$$\mathbb{E}\left[\left(\sum_{k \in \Lambda_j} \left|\hat{\beta}_{j,k} - \beta_{j,k}\right|\frac{\left|\beta_{j,k}\right|}{t_n}\left|\psi_{j,k}(x_0)\right|\right)^p\right]$$

$$= \mathbb{E}\left[\left(\sum_{k \in \Lambda_j} \left|\hat{\beta}_{j,k} - \beta_{j,k}\right|\frac{\left|\beta_{j,k}\right|^{1/p}}{t_n^{1/p}}\left|\psi_{j,k}(x_0)\right|^{1/p}\frac{\left|\beta_{j,k}\right|^{1/p'}}{t_n^{1/p'}}\left|\psi_{j,k}(x_0)\right|^{1/p'}\right)^p\right]$$

$$\le \mathbb{E}\left[\sum_{k \in \Lambda_j} \left|\hat{\beta}_{j,k} - \beta_{j,k}\right|^p\frac{\left|\beta_{j,k}\right|}{t_n}\left|\psi_{j,k}(x_0)\right|\right]\left(\sum_{k \in \Lambda_j}\frac{\left|\beta_{j,k}\right|}{t_n}\left|\psi_{j,k}(x_0)\right|\right)^{p/p'}$$

$$\lesssim n^{-p/2} t_n^{-p} 2^{-jps} \lesssim (\ln n)^{-\frac{p}{2}} 2^{-jps}.$$

Hence,

$$\left[\sum_{j=j'+1}^{j_1} (\ln n)^{-\frac{p}{2}} 2^{-jps}\right]^{1/p} \lesssim (\ln n)^{-\frac{1}{2}} 2^{-j's}. \tag{27}$$

Combing (26), (27) and $2^{j'} \sim \left(\frac{n}{\ln n}\right)^{1/(2s+1)}$, one gets

$$Q_2 \lesssim \left(\frac{2^{j'}}{n}\right)^{1/2} + (\ln n)^{-\frac{1}{2}} 2^{-j's} \lesssim \left(\frac{\ln n}{n}\right)^{s/(2s+1)}. \tag{28}$$

- For $Q_3$. Note that

$$Q_3 = \left(\sum_{j=j_*}^{j'} + \sum_{j=j'+1}^{j_1}\right)\sum_{k \in \Lambda_j}\left|\beta_{j,k}\right| I_{\{|\beta_{j,k}| \le 2\kappa t_n\}}\left|\psi_{j,k}(x_0)\right| =: Q_{31} + Q_{32}.$$

It is easy to show that

$$
\begin{aligned}
Q_{31} &= \sum_{j=j_*}^{j'} \sum_{k \in \Lambda_j} \left| \beta_{j,k} \right| I_{\left\{ |\beta_{j,k}| \le 2\kappa t_n \right\}} \left| \psi_{j,k}(x_0) \right| \\
&\lesssim \sum_{j=j_*}^{j'} \sum_{k \in \Lambda_j} \left| \beta_{j,k} \right| \frac{2\kappa t_n}{\left| \beta_{j,k} \right|} \left| \psi_{j,k}(x_0) \right| \lesssim \sum_{j=j_*}^{j'} 2^{\frac{j}{2}} t_n \lesssim 2^{\frac{j'}{2}} \sqrt{\frac{\ln n}{n}}.
\end{aligned}
\tag{29}
$$

In addition,

$$
\begin{aligned}
Q_{32} &= \sum_{j=j'+1}^{j_1} \sum_{k \in \Lambda_j} \left| \beta_{j,k} \right| I_{\left\{ |\beta_{j,k}| \le 2\kappa t_n \right\}} \left| \psi_{j,k}(x_0) \right| \\
&\lesssim \sum_{j=j'+1}^{j_1} \sum_{k \in \Lambda_j} \left| \beta_{j,k} \psi_{j,k}(x_0) \right| \lesssim \sum_{j=j'+1}^{j_1} 2^{-js} \lesssim 2^{-j's}.
\end{aligned}
\tag{30}
$$

Then according to (29), (30) and $2^{j'} \sim \left( \frac{n}{\ln n} \right)^{1/(2s+1)}$, one can obtain

$$
Q_3 \lesssim 2^{\frac{j'}{2}} \sqrt{\frac{\ln n}{n}} + 2^{-j's} \lesssim \left( \frac{\ln n}{n} \right)^{s/(2s+1)}.
\tag{31}
$$

Furthermore, together with (25) and (28), this yields

$$
\begin{aligned}
Q &\lesssim (\ln n)^{1-\frac{1}{p}} \left( n^{-\frac{s}{2s+1}} + \left( \frac{\ln n}{n} \right)^{s/(2s+1)} + \left( \frac{\ln n}{n} \right)^{s/(2s+1)} \right) \\
&\lesssim (\ln n)^{1-\frac{1}{p}} \left( \frac{\ln n}{n} \right)^{s/(2s+1)}.
\end{aligned}
\tag{32}
$$

Finally, it follows from (20), (21) and (32) that

$$
\sup_{r \in H^s(\Omega_{x_0})} \left\{ \mathrm{E} \left[ |\hat{r}_n^{non}(x_0) - r(x_0)|^p \right] \right\}^{\frac{1}{p}} \lesssim (\ln n)^{1-\frac{1}{p}} \left( \frac{\ln n}{n} \right)^{s/(2s+1)},
$$

which completes the proof of Theorem 2. □

## 5. Conclusions

This paper studies the pointwise estimations of an unknown function in a regression model with multiplicative and additive noise. Under some different assumptions, linear and nonlinear wavelet estimators are constructed. It is clear that those wavelet estimators have diverse forms with different conditions. The convergence rates over the pointwise risk of two wavelet estimators are proposed by Theorems 1 and 2. It should be pointed out that the linear and nonlinear wavelet estimators can all obtain the optimal convergence rate of pointwise nonparametric estimation. More importantly, the nonlinear wavelet estimator is adaptive. In other words, the conclusions of asymptotic and theoretical performance are clear in this paper. However, it is a difficult problem to give numerical experiments, which need more investigations and new skills. We will study it in the future.

**Author Contributions:** Writing—original draft, J.K. and Q.H.; Writing—review and editing, H.G. All authors have read and agreed to the published version of the manuscript.

**Funding:** Junke Kou is supported by the National Natural Science Foundation of China (12001133) and Guangxi Natural Science Foundation (2019GXNSFFA245012). Huijun Guo is supported by the National Natural Science Foundation of China (12001132), and Guangxi Colleges and Universities Key Laboratory of Data Analysis and Computation.

**Institutional Review Board Statement:** Not applicable.

**Informed Consent Statement:** Not applicable.

**Data Availability Statement:** Not applicable.

**Acknowledgments:** The authors would like to thank the anonymous reviewers for their helpful comments.

**Conflicts of Interest:** The authors state that there is no conflict of interest.

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
