# Peer review of "Pointwise Wavelet Estimations for a Regression Model in Local Hölder Space"

_axioms, doi:10.3390/axioms11090466_

Round 1

Reviewer 1 Report

The work is interesting and broadens the scope of knowledge in the field discussed in it. It is written very carefully. I managed to find a bit of syntax errors (I did not find the substantive one). All my comments are in the attached pdf file (green - spots, pink - comments).

Author Response

Dear Reviewers,

First of all, we would like to express our gratitude for your valuable and detailed comments. The manuscript has been thoroughly modified and corrected according to the comments. We hope that the present version will match your expectations. The corrections are given as follows.

For the comments,some syntax errors and misprints are revised.

Best Regards!

Junke Kou, Qinmei Huang and Huijun Guo

Reviewer 2 Report

The paper presents an interesting topic in the field of mathematics and applied mathematics in science.

The work is based on a regression model for which two wavelet estimators were proposed and the convergence rates under mean integrated squared error over Besov space were discussed. The authors of this paper, take this idea and construct two new wavelet estimators, and the convergence rates over pointwise error of the wavelet estimators in local Holder space, are taken into account. These wavelet estimators can achieve the optimal convergence rate under pointwise error.

The introduction provides sufficient background; Introduction focuses clearly on the main point.

The explanations and mathematical demonstrations are very well presented.

There are several points to be improved:

1.      Numerical experiments (or numerical approximations to the convergence rate) can be provided to demonstrate the theoretical results? For example, I think it could be presented for the convergence rate of linear wavelet estimator or for nonlinear wavelet estimator with hard thresholding method.

2.      Even if the traditional paper format (introduction, materials and methods, results, discussion and conclusions) is not exactly used, I think it would be useful to introduce a section in which at least one numerical experiment is presented and the key points are shown (the obtained results based on the theoretical information demonstrated in this paper).

Author Response

Dear Reviewers,

First of all, we would like to express our gratitude for your valuable and detailed comments. The manuscript has been thoroughly modified and corrected according to the comments. We hope that the present version will match your expectations. The corrections are given as follows.

For the comments, we add a conclusion to explain it on page 14. This paper considers the pointwise estimations of linear and nonlinear wavelet estimators for a regression model with multiplicative and additive noise. It turns out that those linear and nonlinear wavelet estimators all can obtain the optimal convergence rate of pointwise nonparametric problem. What’s more, the nonlinear estimator is adaptive. The conclusions of asymptotic and theoretical performance are clear in this paper. However, the numerical experiments demand more investigations and skills, we do not actually have and that we leave for future work.

Best Regards!

Junke Kou, Qinmei Huang and Huijun Guo

Round 2

Reviewer 1 Report

After the introduced corrections, in my opinion, the work is ready for printing, which I recommend hereby.

Reviewer 2 Report

Dear authors,

I hereby confirm receipt of the revised version of your paper.

In this form, the paper can be published in this journal.